# A Review of the Methodology of Analyzing Aflatoxin and Fumonisin in Single Corn Kernels and the Potential Impacts of These Methods on Food Security

**DOI:** 10.3390/foods9030297

**Published:** 2020-03-05

**Authors:** Ruben A. Chavez, Xianbin Cheng, Matthew J. Stasiewicz

**Affiliations:** Department of Food Science and Human Nutrition. University of Illinois at Urbana-Champaign. 905 S Goodwin Ave., Urbana, IL 61801, USA; rubenc2@illinois.edu (R.A.C.); xianbin2@illinois.edu (X.C.)

**Keywords:** mycotoxin, global food safety, mycotoxin prevention, mycotoxin surveillance, single kernel, aflatoxin, fumonisin, corn

## Abstract

Current detection methods for contamination of aflatoxin and fumonisin used in the corn industry are based on bulk level. However, literature demonstrates that contamination of these mycotoxins is highly skewed and bulk samples do not always represent accurately the overall contamination in a batch of corn. Single kernel analysis can provide an insightful level of analysis of the contamination of aflatoxin and fumonisin, as well as suggest a possible remediation to the skewness present in bulk detection. Current literature describes analytical methods capable of detecting aflatoxin and fumonisin at a single kernel level, such as liquid chromatography, fluorescence imaging, and reflectance imaging. These methods could provide tools to classify mycotoxin contaminated kernels and study potential co-occurrence of aflatoxin and fumonisin. Analysis at a single kernel level could provide a solution to the skewness present in mycotoxin contamination detection and offer improved remediation methods through sorting that could impact food security and management of food waste.

## 1. Introduction

Mycotoxins are a regular problem in food and feed products, and the occurrence of these toxins generate a concern in the food industry, with aflatoxin and fumonisin creating a major concern in corn. Aflatoxins (AF) are a group of a secondary metabolite produced by some strains of *Aspergillus flavus*, *A. parasiticus* and *A. nomius* [1]. There are four main aflatoxins: aflatoxin B1 (AFB1), aflatoxin B2 (AFB2), aflatoxin G1 (AFG1) and aflatoxin G2 (AFG2); with AFB1 being the most prevalent in contamination of corn [2,3,4]. Consumption of AF impairs the function of liver cells which facilitates cancer development, creating hepatocellular carcinoma [5]. Fumonisin (FM) is another mycotoxin that naturally occurs in corn. FM is a class of secondary fungal metabolites produced by *Fusarium verticillioides* and *F. proliferatum*. By 2010, more than 28 different isomer of FM have been discovered [6]. Fumonisin B1 (FB1) accounts for 60% or more of the fumonisin present in maize [7]. FM can cause esophageal cancer and defects in the neural system [8]. Due to their health risk associated with consumption of AF and FM, organizations such as the United States Department of Agriculture (USDA), the European Committee for Standardization (CEN) and International Organization for Standardization (ISO) have established analytical sampling methods, focused on bulk testing, for the detection and quantification of these toxins, such as high-performance liquid chromatography (HPLC), or enzyme-linked immunosorbent assay (ELISA).

Unfortunately, research conducted in detection of mycotoxins has shown that AF and FM contamination occurrence distribution is irregular and the bulk sampling of a batch of corn does not always represent accurately the contamination of AF and FM [9,10]. While the majority of corn kernels are uncontaminated or low in mycotoxins, there are a few hotspots where kernels are highly contaminated [1,10,11]. Since mycotoxin contamination occurrence is irregular and unpredictable it can affect the overall accuracy of bulk analysis, creating two types of statistical errors where samples can falsely test over a regulatory level and be rejected or they can falsely test under a regulatory level with a production unit wrongly accepted [9]. These ambiguities in sampling error within bulk analysis pose a threat to food safety.

Such a dilemma could be resolved by the usage of a single kernel analysis of AF and FM contamination; having single kernel detection would increase the understanding of the range of concentrations in a bulk of grain. Previous studies made on single kernel analysis of mycotoxins that encompass the utilization of wet chemistry, spectral methods, or mathematical algorithms propose potential solutions to the skewedness present in AF and FM contamination in corn.

There are several methods for AF/FM detection in single corn kernels, such as liquid chromatography, fluorescence imaging, infrared imaging, and enzyme-linked immunosorbent assay. Further implementation of these detection methods has yielded sampling models or sorting mechanisms capable to minimize the impact associated with the distribution of mycotoxin contamination and improve classification of contaminated kernels [1]. As further discussed in this review paper, single kernel analysis could improve the risk assessment of AF and FM. It could provide more accurate analysis of the contamination of corn samples, enable a sorting system that can classify contaminated corn, and improve multiple mycotoxin detection by reducing any correlations between co-occurrence of mycotoxins. This paper will first address methods to analyze AF and FM in single corn kernels, both historic methods and current techniques including liquid chromatography, fluorescence imaging, and infrared imaging. Secondly, this paper will review basic scientific findings, including classification and correlation of AF and FM. Finally, this paper will discuss the impacts of these classification and correlation findings on potential food safety applications that could generate sorting mechanisms, improve risk assessments, and reduce food waste.

## 2. Techniques to Analyze Mycotoxins in Single Kernels

To our knowledge, the first paper to report identification of mycotoxin on a single corn kernel level was Lee, Lillihoj, and Kwolek in 1980. AF levels were evaluated by chloroform extraction assay and correlated to bright greenish-yellow (BGY) fluorescence. The experiment showed 85% of contaminated kernels that were also BGY fluorescent, had a range from 100 to 8000 ng/g total AF and were often adjacent to uncontaminated kernels; this study demonstrated that non-destructive, spectral methods may be useful tools to identify AF present single kernels. BGY fluorescence was one of the first methods to screen for AF contamination in kernels representing a bulk lot. By examining the kernels for yellow-green fluorescence under a black light, scientists were able to indicate possible presence of AF in them [12]. In 1998, FM was identified as a mycotoxin present in corn. A group of South African researchers were able to characterize the FM as diesters of tricarballylic acid and C22-amino-alcahols, and linked it to esophageal cancer [13,14]. Liquid chromatography was used as one of the first methods to identify and quantify the FM contamination [14]. These first publications, about AF and FM presence in corn, created a foundation for new research studies. Later publications focus on bulk corn testing of co-occurrence of mycotoxins and try to develop procedures that could minimize the contamination of mycotoxin in corn [15,16,17]. Currently, bulk level identification of mycotoxins is used as the standard measurement for AF and FM detection in corn and other commodities. However, it is known that mycotoxin contamination is highly skewed and the bulk level might not represent accurately the contaminated kernels within each sample [10]. Data obtained from single kernel mycotoxin detection methods, that involve a non-destructive sample path, could help the agriculture industry to develop models that are capable of detecting accurately the kernels that are contaminated with mycotoxins. Table 1 presents a summary of the studies conducted to detect mycotoxins at a single corn level, with the main categories of analytical techniques being: liquid chromatography, fluorescence imaging, infrared imaging, and enzyme-linked immunosorbent assay.

### 2.1. Liquid Chromatography

One of the standard methods for identification and quantification of mycotoxin is liquid chromatography [18]. There are several variations in liquid chromatography methods that have been used for AF and FM detection, such as thin layer chromatography (TLC), high performance liquid chromatography (HPLC) and liquid chromatography tandem mass spectrometry (LC-MS/MS) [19]. The first experiments that tried to quantify the AF contamination in corn relied on simpler chloroform extractions with TLC [12,20]. Each variant of liquid chromatography has its own variance and sensitivity of mycotoxin detection, and the desired outcome of the analysis will decide the type of liquid chromatography utilized for mycotoxin detection [21].

Research conducted in AF and FM contamination in corn at single kernel level utilizes a variant of liquid chromatography to quantify the concentration of mycotoxin in samples in combination with other methods to accurately identify the type of mycotoxins present in the analyzed sample [10,19,21,22]. One of the first reports that focused on single kernel identification was made by Mogensen et al., 2011. It analyzed the total FM concentration in 400 corn kernels by using LC-MS/MS, the results detected 15% of the kernels to be contaminated with FM [23]. Liquid Chromatography has a range of analytical characteristics that can be utilized for mycotoxin analysis. In combination with other analytical instruments, such as mass spectrometry or gas chromatography, it can yield results that demonstrate different stereoisomer of AF or FM [24]. HPLC can present a procedure with high selectivity, and simultaneous quantification of mycotoxin [19,24,25]. Nonetheless, spectral libraries are needed, meticulous sample preparation is required, and major costs are associated with the chemical and personnel required to operate through this method.

### 2.2. Fluorescence Imaging

Bright green-yellow (BGY) fluorescence, or black light testing, is one of the first methods to screen for AF through fluorescence. Corn is viewed under an ultraviolet light (365 nm) and may emit greenish-yellow fluorescence correlated with AF contamination in corn kernels; nonetheless, BGY fluorescence does not necessarily represent actual contamination of aflatoxin, instead BGY fluorescence is associated with fluorescence of kojic acid, which is also a secondary fungal metabolite that is commonly produced by many species of *Aspergillus*; some of which also produce AF as a secondary metabolite.

As research progressed in AF bulk identification through fluorescence, black light testing became one of the standard methods for inspection of AF contamination, mentioned in GIPSA inspection manual Standards. Other studies started to focus on the single kernel aspect of fluorescence [26]. Further analysis of the spectra emitted under UV conditions demonstrated that AF has excitation peaks at certain wavelengths [27]. AFB1 has excitation peaks at wavelengths 223, 265 and 362 nm; likewise, AFB2 has excitation peaks at 222, 265 and 363 nm; demonstrating that specific wavelengths in fluorescence spectra can be utilized to detect AF contamination in corn at single kernel level.

One variant of UV fluorescence analysis of mycotoxins is the use of hyperspectral imaging of UV fluorescence to analyze AF contamination. Hyperspectral imaging incorporates spatial imaging techniques and spectroscopy to provide better information about the spatial and spectral information obtained from the sample tested [28]. Hyperspectral fluorescence has been used in the detection of total AF contaminated products. It can present a rapid nondestructive screening detection for corn that is contaminated with AF. A research study that utilized fluorescence emission at a single kernel level with stationary samples of corn kernels was performed, and data was taken with a fluorescence hyperspectral imaging system under ultraviolet light. These data demonstrated that AF contaminated kernels have a fluorescence peak shift toward longer wavelength, and a lower fluorescence peak magnitude [26]. This method could present an alternative approach for AF detection in contaminated kernels. Further analysis in hyperspectral imaging of UV fluorescence in contaminated corn demonstrates improved detection for AF in corn. Zhu et al., 2016, demonstrates that, under UV excitation, kernels with higher contamination levels had fluorescence peaks at longer wavelengths with lower intensity. It was shown that spectral data from kernels with total AF concentration lower than 20 ppb had a difference in peak shift of 17 nm in comparison to the kernels that were above 100 ppb [28]. Due to the potential of hyperspectral fluorescence identification of AF, other methods have been tested that demonstrated possible detection of aflatoxin at a single corn level. High-speed dual-camera systems have been tested by using fluorescence principles to detect total AF contamination. A study was performed using this technology by screening AF contamination in corn using multispectral fluorescence imaging. The method consisted in two cameras that simultaneously captured a single-band fluorescence image at 436 or 532 nm and used commercial samples that were inoculated. Through chemometric analysis of the spectral data, the high-speed dual-camera system was able to distinguish correctly AF inoculated samples from non-inoculated corn.

Research about fluorescence analysis of FM in corn at a single kernel level is limited [28]. Most of the detection of FM in corn kernel encompass wet chemistry or reflectance in infrared regions. Silva et al. 2009, utilized fluorescence in combination with liquid chromatography and mass spectrometry to detect FB1 and FB2 in corn based food [25]. Moreover, other aspects of fluorescence are used to detect and quantify FB1; one of them being fluorescence polarization (FP). The FP process could potentially be utilized as a better screening method in monoclonal antibodies analyses that are capable to detect and quantify FM; however current enzyme-linked immunosorbent assays (ELISA) tend to be more sensitive and more research is needed.

Overall, fluorescence imaging represents an alternate non-invasive method to detect mycotoxins contamination in commodities and presents methods that have a lower cost per sample than other current analytical methods used for the same purpose, but it still has limitations that increase the difficulty of its application. One of the major limitations of fluorescence emission is the low signal level that could present errors at the moment of recording the maximum fluorescence intensities and emissions [19]. Additionally, internal contamination of instruments is possible and there is significant cross-contamination when screening all corn kernels in a large lot [19,26]. In addition, corn commodity tends to contain other natural products that have natural fluorescence characteristics that can overlap with the signal of the kernels analyzed.

### 2.3. Infrared Imaging

Infrared spectroscopy encompasses a broad spectrum of wavelength (800 nm–1,000,000 nm). Trying to use it as method to detect relatively small molecules such as mycotoxins (MW = 700) generates drawbacks and possible limitations, such as low sensitivity in the signal emitted by mycotoxins [29,30]. Despite the disadvantages, studies have shown that the near infrared region (NIR) (800 nm–2500 nm) is capable of differentiating kernels containing high (>100 ppb) or low (<10 ppb) levels of total AF, as well as detecting kernels containing high (>10 ppm) or relatively low (<10 ppm) levels of total FM [31,32]. Current single kernel analysis methods with NIR have two main modes of data collection: reflectance, and transmittance. The selection of the method will vary on the type of the samples and the constituents that will be tested [18,33,34].

Furthermore, spectral data obtained from NIR tends to be complex, and overlapping as well as weak absorptions bands tend to generate complex spectral results, creating the need for analytical tools to solve this problem. Chemometrics are commonly used as a tool for spectral analysis of NIR in mycotoxin detection. One study uses short wave infrared hyperspectral imaging (SWIR) to detect AF in corn, in combination with a chemometric system, principal component analysis (PCA). The spectral results show specific range wavelengths associated with the chemical structure of AFB1 [31]. Kandpal et al., have done research on how efficient this technique can be in terms of categorization of infected corn kernels by contrasting the efficiency of SWIR with and without the PLS-DA analysis [35]. The study analyzes the spectral data of 585 corn kernels with a variety of concentrations and tests the detection accuracy of AFB1 using SWIR with standard samples. By using SWIR with partial least squares discriminant analysis (PLS-DA), they were able to have an accuracy of 96.9% for AF detection, in a contrast to the 86.6% when only SWIR is used [35]. SWIR can be an inexpensive analytical method for rapid detection for AF in combination of PLS-DA. Pearson et al. has conducted research in NIR detection for AF and was able to demonstrate that it is capable of evaluating AF contamination in corn. Pearson utilizes both reflectance and transmittance over a spectral range of 500 to 950 nm and 550 to 1700 nm to collect spectral data of artificially inoculated corn kernels. By utilizing PLS-DA in the spectral results, 95% of contaminated kernels were classified as high (>100 ppb) or low (<10 ppb) in total AF contamination. In further experiments, Pearson was able to develop and test a high-speed dual-wavelength sorter that was able to detect total AF and FM contamination in kernels based on reflectance with a higher spectrum between 400 and 1700 nm [36,37]. Further analysis under mycotoxin contamination helped improve AF and FM detection through NIR spectra. Stasiewicz et al. utilizes naturally AF and FM contaminated corn kernels obtained from Kenya to analyze them through Vis-NIR spectroscopy. The study utilizes reflectance values at nine distinct wavelengths between 470 and 1550 nm, and analyze the spectra through PLS-DA. Total AF and FM were identified with a sensitivity (Sn) of 0.77 and specificity (Sp) of 0.83.

Other studies that focus on single kernel analysis utilizes instrumentation that enhances the NIR analysis for mycotoxins detection. Fourier transform near infrared (FT-NIR) spectroscopy can be used as a rapid detection method for FM. FT differs from other spectral methods in that all of the resolution elements for a spectrum are measured simultaneously. In addition, FT instrumentation improves resolving power, wavelength reproducibility, and greater signal to noise ratio [38]. Research done on 168 samples of corn collected from different parts of Italy demonstrates that FT-NIR spectroscopy is an easier and faster method to detect FB1 and FB2 in corn compared to traditional techniques such as HPLC and ELISA [11]. The results obtained in the FT-NIR were used to create a model that predicts FM concentration. Statistical software was used to test the model and none of the data collected by FT-NIR fell in the rejection region [11]. FT-NIR gives the possibility to obtain accurate spectral results faster rate than regular NIR reflectance.

Further research conducted in single kernel mycotoxin detection has shown that the use of visible light, ultraviolet light, or both, in conjunction with NIR, can improve the detection of AF. A study that uses visible/near-infrared, in combination with factorial discriminant analysis, was able to detect AFB1 with a validation accuracy of 96% for the samples tested in maize kernels [33]. Tao, Feifei, et al. 2018 study uses visible-near-infrared (Vis-NIR) spectroscopy to detect contamination of total AF in corn kernels. In combination with competitive adaptive reweighted sampling (CARS) with partial least-squares discriminant analysis (PLS-DA) models, the research was able to achieve an overall accuracy of detection of 87% [18]. Overall, NIR results demonstrate an alternative method to detect AF and FM in kernels, in combination with correct statistical methods that are capable to analyze the spectral results correctly.

### 2.4. Enzyme-Linked Immunosorbent Assay

Enzyme-linked immunosorbent assay (ELISA) is another category of mycotoxin analysis method developed for bulk analysis where, in this context, an aqueous extract of mycotoxin from single corn kernels would be delivered to a surface coated with AF or FM antibodies and levels of toxin quantified based on comparison to surfaces exposed to standard toxin concentrations. These methods are generally rapid, inexpensive, and relatively easy to use compared to reference chromatography methods, with acceptably low limits of detection [19]. Antibodies used for ELISA have cross reactivity to multiple mycotoxins, such as for AFB1, AFB2, AFG1, and AFG2, therefore making them more appropriate for total AF or FM quantification than specific toxins, e.g., AFB1, which would have affected detection thresholds and overestimation of results [39]. Yet, some immunochemical methods can have a binding specificity specifically to AFB1 [40]. In the context of this review of single kernel AF and FM detection, ELISA is generally used in studies developing non-destructive fluorescence or infrared imaging techniques as a reference method to provide the true status of a training set of single kernels [1,10,19,41].

## 3. Outcomes of Single Kernel Mycotoxin Analysis

### 3.1. Classification of Aflatoxin and Fumonisin on a Single Kernel Level

Spectral data obtained from fluorescence or infrared imaging offer a building block for classification models that could predict mycotoxin contamination. However, raw data require pre-processing steps that increase signal-to-noise ratio and strengthen some of the spectral features that may facilitate classification. Several common pre-processing techniques have been utilized in the literature, such as background subtraction [1], Savitzky–Golay filtering, standard normal variate (SNV) transformation or normalization, and principal component analysis (PCA) [28]. Some of these techniques may be used in combination to transform spectral data with the goal of enhancing model accuracy. Then, data are analyzed further through machine learning algorithms and each of them has a different set of strengths and weaknesses. Based on the literature, while discriminant analysis (DA) and its variants have been predominantly employed in AF detection studies, other algorithms such as k-nearest neighbors (KNN), support vector machine (SVM), and random forest (RF) are gaining popularity [10,26,28,37]. With the pre-processed spectral data and a chosen algorithm, one can proceed to train a model with known samples and test it with unknown samples. It is conventional to split the spectral data into two portions of differing size. The larger portion is used as a training dataset where spectral features and AF status for each kernel are both present and the smaller portion is treated as a test dataset where only spectral features are present. Once the model is built on the training dataset, it will be used to predict each kernel’s AF status in the test dataset [42]. The predicted results are compared with the actual AF status and evaluation metrics such as overall accuracy, sensitivity, and specificity will be produced [43]. These metrics are indicative of the model’s ability to correctly classify unknown corn kernels. So far, there is no standardization on usage of model evaluation metrics but sensitivity and specificity tend to be preferable as they are more informative than overall accuracy for describing model performance.

Research conducted in the area has proven that detection or classification methods at a single kernel level, in conjunction with data analysis, are capable of creating detection models that identify contaminated kernels. For example, one study that utilizes transmittance (550–950 nm) and reflectance (550–1700 nm), in combination with DA and PLS to analyze contaminated corn kernels, can generate an algorithm capable of classifying kernels with high levels (>100 ppb) or low levels (<10 ppb) of total AF contamination, with a 95% accuracy [36]. Further research regarding this topic utilize Vis-NIR reflectance (500–1700 nm) and DA to analyze total AF contamination in corn kernels; model accuracy results yielded a 98% for classification of AF contaminated kernels (>100 ppb) and uncontaminated kernels [37]. In addition, hyperspectral fluorescence imaging, in combination with LS-SVM, KNN or both, can yield models that could classify correctly AF. One study utilizes hyperspectral fluorescence (399–701 nm), with KNN algorithm, to detect and classify AF contamination in corn. After spectral analysis, results yield 84% sensitivity and 95% specificity, with a threshold of 20 ppb of total AF [44]. Moreover, additional research was conducted utilizing the same spectral range, employing LS-SVM and KNN to generate the classification model. Model accuracies displayed a 91% sensitivity and 96% specificity at a classification threshold at 100 ppb, as well as 89% sensitivity and 96% specificity at a classification threshold of 20 ppb of total AF [28]. Nonetheless, most of literature that focus on this area utilize stationary kernels to obtain the required data for classification methods.

Classification models analyzing kernels in motion could create practical results that could be utilized in the corn industry. To our knowledge, there are only two studies that utilize in motion single kernel analysis to generate spectra for classification model. The first study was performed in Kenya, and demonstrated that NIR spectroscopy of mycotoxins in moving kernels in combination with LDA, was capable of identify AF with a 77% sensitivity and an 83% specificity [10]. By further analyzing the spectral data through classification algorithms and a multispectral sorter, the results were able to develop a sorting mechanism for AF and FM contaminated kernels [10]. The second study utilizes UV-Vis-NIR (304–1086 nm) with RF to classify AF contaminated kernels. Accuracies of the model yielded 86% sensitivity and 97% specificity at a classification threshold of 20 ppb of total AF [1]. Existing literature lacks a focus of single kernel classification and sorting. Currently, studies that create a modeling system for classification and sorting at a single kernel level use the UV, visible, and IR wavelength to demonstrate proper classification techniques through modeling systems summarized in Table 1. Building a classification model is usually the final and most critical process in a spectral-based nondestructive AF detection project. It is a pivotal point that connects the spectral pattern of sample with its contamination status determined by chemical assay. A well-trained and optimized model should be capable of generalizing the relationship between spectral and chemical information from known samples and distinguishing contaminated kernels from uncontaminated ones in unknown samples. Further research conducted in single kernel mycotoxin detection has shown that the use of visible light, ultraviolet light, or both, in conjunction with NIR, can improve the detection of AF. A study that uses visible/near-infrared, in combination with factorial discriminant analysis, was able to detect AFB1 with validation accuracies between 67.5% and 100% [33].

### 3.2. Relationship between Aflatoxin and Fumonisin on a Single Kernel Level

Many biological factors are related to toxin accumulation in maize, and the analysis of all the factors that are related to mycotoxin contamination are complex. Contamination occur once a fungus that is capable of producing mycotoxin infects a kernel, and single kernel contamination tends to occur by different causes. Many types of fungi are able to produce several types of mycotoxin, and co-occurrence of mycotoxins is a real issue in food and feed [45]. Once infection occurs, field conditions or storage conditions, such as high moisture content, high relative humidity, and mild temperatures, increase the proliferation of *Fusarium* and *Aspergillus*, which may produce FM and AF, respectively [46]. These conditions that enhance the proliferation of fungi can yield the co-contamination in corn kernel with more than one type of mycotoxins. Fungi tend to infect kernels at a single kernel level due to insect damage. Mites and insects destroy the outer layer of corn kernels making it difficult to prevent the proliferation of two fungi in the kernel and consequently resulting in two or more toxins [46]. Literature demonstrates that co-occurrence of mycotoxins is very common in bulk measurements even if the sample is contaminated with one mycotoxin, that will not result in avoidance of a second type of mycotoxin contamination. Single kernel analysis can provide a better understanding for AF and FM co-occurrence and provide better mitigation procedures for the reduction of contaminated corn kernels. Nonetheless, there is limited literature that investigates the co-contamination of AF and FM at single kernel level.

Co-contamination is already a current issue in the industry. Studies at the bulk level demonstrate that AF and FM can co-occur in contaminated maize, with mean levels that range between 10 ppb and 44 ppb for AF, and 2 ppm and 5 ppm for FM [47,48,49]. One study performed in the rural areas of Tanzania analyzed the co-occurrence of AF and FM in corn kernels by ultra-high performance liquid chromatography/time-of-flight mass spectrometry (TOFMS) with a QuEChERS-based procedure as sample treatment; in the study, 50% of the samples were contaminated with different types of AF and 43% of all samples were contaminated with FB1 and FB2, and contamination of more than one mycotoxin was observed in 87% of their samples.

In single kernel studies, Borutova et al. study corn kernels for food stock from different regions of Oceania and Asia. The study shows that the contamination of only one type of mycotoxin is very rare and co-contamination of AF and FM is the most common co-occurrence among mycotoxins. A total of 1468 corn kernels were analyzed by HPLC and correlation analysis ($r^{2}$ = 0.85–0.88); the highest correlation between mycotoxins was AFB1 and FB1 [50]. From the results, it was expected to obtain co-occurrence of AF and FM in the same corn kernel if one of the samples is positive in one type of mycotoxin [50]. Likewise, Stasiewicz et al. was able to detect that FM frequently co-occurs with AF in African maize after fluorescence screening under ultraviolet light, with 10% of the samples contaminated with AF and FM [10]. While other studies have demonstrated that a small portion of contaminated kernels could present co-contamination of AF and FM [10,51,52,53,54]. Overall, various studies around the world on mycotoxins co-occurrence demonstrate that contamination with only one type of mycotoxin is common; nonetheless, co-occurrence of more than one mycotoxin is expected in both in bulk analysis and at a single kernel level; particularly when prevalence is high.

## 4. Impact of Mycotoxin Detection on Single Kernel Corn

The major characteristics of single kernel analysis that impact food security are the detection of mycotoxins and the potential to lead to sorting of contaminated kernels. Fluorescence imaging of single kernels, contaminated with AF and FM, has proven to be an effective method to locate contaminated kernels in a production lot. Multiple research articles have observed that kernels with a high concentration in AF are capable to be identified and distinguished from uncontaminated kernels [19,26,27,28]. The detection of AF at a single kernel level through fluorescence has created a new opportunity for mycotoxin detection. A device capable to detect AF contamination through fluorescence was already created to solve the current problem in AF detection, called “Aflagoggles”. The device consists of an ultraviolet light source that is coupled with a light-excluding compartment, and the UV fluorescence is collected to a spectral imaging camera [55]. Peak shifts in the spectral range from 451 nm to 500 nm are used to correlate AF contamination.

Infrared imaging has also proven to be an efficient method for detection and spectral sorting of mycotoxins. Several researches have established that analysis of mycotoxins under NIR in combination with UV, visible light and chemometrics can detect accurately kernels that are highly contaminated with AF and FM [10,11,18,56]. Spectroscopy represents a unique and non-destructive way to detect mycotoxin contamination in maize [19]. Pearson et al. demonstrates the use of a dual wavelength high-speed commercial sorter in AF and FM contaminated kernels, by reducing 81% of AF contamination and 85% of FM contamination reducing in commercial corn samples [37]. Further research in the dual wavelength sorter has showed that it is capable to reduce 46% of AF contamination and 57% of FM contamination.

One of the major issues in mycotoxin analysis is the detection of intermediate level of contaminated corn kernels, improvements in rapid screening for intermediate contamination in corn samples could generate a better alternative for mycotoxin detection. Research in single kernel sorting has proved that reduction of highly AF contaminated kernels can be achieved through spectral methods, and advances in these areas can reduce the sampling error in the detection of intermediate level contamination of AF [1,10]. If high specificity and sensitivity are achieved, the highly contaminated kernels in a grain production lot could be removed accurately and the wholesome ones could be retained for consumption.

Buhler was able to analyze further the hotspot problem of mycotoxins located in maize and the need for a more effective sorting procedure, by developing a LumoVision spectrometer capable of identifying AF through fluorescence hyperspectral imaging. Analysis of the LumoVision spectrometer demonstrates that 90% of AF was reduced through targeted detection and elimination of maize kernels that carry high contamination [57]. Nonetheless, one of the drawbacks of using optical sorting is the performance of the machine even with 80% to 90% bulk reduction, and mycotoxins levels in the accepted fraction of a highly contaminated lot may be above accepted limits [10]. Despite the disadvantages, spectral classification offers a possible solution for AF detection and reduction of contamination. Tools such as the LumoVision spectrometer or the Aflagoggles help managing the mycotoxins contamination present in corn correctly. Developments in future sorting machinery and commercially available sorting methods in single kernel could minimize the risk of AF and FM contamination, and improve food security among consumers. In addition, management of co-contamination can be present by utilizing single kernel sorting techniques. Research articles have shown that single kernel detection can detect multiple mycotoxins [46,50,58]. Single kernel detection could not only provide a solution for a single mycotoxin detection such as AF and FM, as well as improve the food security by reducing the risk factor of co-occurrence of mycotoxins in corn.

## 5. Conclusions

Skewed single kernel mycotoxin distribution can lead to testing error where batches of corn are classified as safe to consume despite having some highly contaminated kernels that can harm consumers; conversely, a few contaminated kernels can cause rejection of an overall acceptable lot just because the sample used for bulk analysis is tested positive. This problem in bulk analysis generates an important opportunity for methods that can mitigate this skewed issue, with sampling single kernel analysis being one of the major solutions. Liquid chromatography, fluorescence imaging, and infrared imaging present analytical methods that can be utilized for single kernel detection. From this review, hyperspectral fluorescence is capable to detect highly contaminated kernels with AF [19,28]. In addition, literature demonstrates that infrared imaging can detect both AF and FM. Research conducted in the NIR region, in combination with chemometric techniques, can demonstrate accurate results in detection of AF and FM [10,35,37]. Further developments in the analysis of the NIR demonstrate that enhancements in the instrumentation or wavelength, such as FT-NIR, SWIR or UV-Vis-NIR, can detect contamination of AF and FM with improved accuracies [11,18,33]. The spectral data obtained from infrared imaging or fluorescence imaging present a nondestructive AF and FM detection project.

Furthermore, spectral data obtained from fluorescence imaging and infrared imaging can be utilized to generate classification algorithms or sorting mechanism that could generate intrinsic solutions for the skewness present in mycotoxin contamination. Classification models in single kernel analysis already demonstrate classification accuracies above 80% [1,10,26,28,33,36,37], and sorting mechanisms in single kernel level, such as LumoVison spectrometer or Aflagoggle, are capable of reducing AF and FM contamination in a batch of corn [10,55,56,57]. In addition, co-occurrence of mycotoxins can be addressed at single kernel level, where methods that are focused on the detection of one type of mycotoxin can remediate possible co-contamination in a single kernel of corn.

Overall, all the characteristics that are present in single kernel analysis of mycotoxins can provide a potential solution to the problem of bulk analysis. The single kernel assays performed with spectral techniques, modeling analysis, and sorting mechanism can be utilized further for skewness remediation, with NIR being the major contributor to this solution. Single kernel sorting can potentially reduce food waste and improve sorting tools for contaminated samples, as literature demonstrates, it is also capable of detecting multiple types of AF reducing even further the risk of obtaining contaminated corn with mycotoxins; while classification models give the user a prediction of the status of contamination. Although it has been reported that approximately 25% of the worldwide crops are probably contaminated with mycotoxins, it is very challenging to estimate the costs of misclassification of uncontaminated kernels [59]. Nevertheless, single kernel analysis presents a major solution to the current skewness present in bulk analysis of AF and FM in corn. As of date, literature focused on the single kernel area is limited and further comprehension in natural contamination of AF and FM is required to demonstrate a conclusive solution for bulk analysis that could improve the food security from corn.

## Figures and Tables

**Table 1 foods-09-00297-t001:** Summary of research studies that report single corn kernel aflatoxin and fumonisin contamination detection using different analytical methods.

Analytical Method used	Mycotoxin tested	Contaminated corn source	Kernel Motion State	Measurement Type	Spectral Analysis method*	Classification Accuracies and Major Results	Reference
Liquid Chromatography	Total Fumonisin	Contaminated and uncontaminated samples obtained from farmers	-	Tandem mass spectrometry	-	39% of kernels (155/400) were contaminated with 1.84–1428 mg/kg fumonisin. Only 4% were above 100 ppb fumonisin and removal of these kernel reduced average fumonisin content by 71%.	[22]
Fluorescence Imaging	Fumonisin FB_1_	In field inoculated samples	-	Fluorescence Polarization	-	Fumonisin concentration was correlated with fluorescence (r^2^ = 0.85–0.88).	[29]
Fluorescence Imaging/Liquid Chromatography	Fumonisin FB_1_, FB_2_	Obtained from commercial sources	-	Fluorescence detection (FD), mass spectrometry	-	Data validation method reproducibility ≤ 15.9% and recovery 78–110%.	[24]
Fluorescence Imaging	Total Aflatoxin	In field inoculated samples	Stationary	Fluorescence emission	Linear regression	84% classification accuracy at threshold of 20ppb, and 86% at classification threshold of 100ppb.	[26]
Fluorescence/Reflectance Imaging	Total Aflatoxin	In field Inoculated samples	Stationary	Fluorescence, reflectance visible near-infrared (VNIR)	KNN	84% sensitivity and 96% specificity for classification model at a threshold of 20 ppb.	[42]
Fluorescence/Reflectance Imaging	Total Aflatoxin	In Field inoculated samples	Stationary	Fluorescence, reflectance visible near-infrared (VNIR)	PCA, LS-SVM, KNN	Threshold values of 20 and 100 ppb were used. Classification models: 92% sensitivity and 96% at threshold of 100 ppb; 89% sensitivity and 96% specificity threshold of 20 ppb.	[28]
Fluorescence Imaging	Total Aflatoxin	Artificially inoculated kernels from commercial samples	Stationary	Dual-camera multispectral fluorescence	NDFI	Contamination levels were 0.011 to 20 ppb. Screening of contaminated samples demonstrated a high sensitivity (0.987) and high specificity (0.96) at threshold of 20 ppb	[27]
Infrared Imaging	Total Aflatoxin	In field inoculation	Stationary	Reflectance and Transmittance spectra	PLS-DA	>95% accuracy for classifying kernels with >100ppb or <10ppb.	[37]
Infrared Imaging	Total Aflatoxin, Total Fumonisin	In field inoculation	Stationary	High speed dual-wavelength Reflectance	FWHM	Absorbance at 750 and 1200 nm correctly identify >99% of kernels. 98% accuracy for classifying kernels with >100ppb or uncontaminated.	[38]
Infrared Imaging	Fumonisin FB_1_, FB_2_	Natural contamination form local farmers	-	Fourier transform near infrared spectroscopy	PLS	Coefficients of correlation, root mean square error and standard error of calibration were 0.964, 0.630 and 0.632, respectively	[10]
Infrared Imaging	Aflatoxin AFB_1_	Artificial inoculation from commercial samples	Stationary	Short wave infrared hyperspectral imaging	PLS-DA	Yellow, white, and purple corn were scanned. Classification between kernels < 10 ppb and > 1000 ppb was achieved with an accuracy of 97%.	[36]
Infrared Imaging	Total Aflatoxin, Fumonisin	Natural contamination from local farmers	In motion	Infrared, Visible, and Ultraviolet Reflectance	LDA, RF, SVM	Skewed distribution of contamination. Spectrometer capable to classify contamination (sensitivity 77%, specificity 83%) and sort at a lower cost.	[9]
Infrared Imaging	Aflatoxin AFB_1_	In field inoculation	Stationary	Short wave infrared hyperspectral imaging	PCA, SVM	11% of the kernels (13/120) were > 2000 ppb. Classification accuracies were 84% and 83% for calibration and validation set, respectively, at thresholds of 20 ppb and 100 ppb.	[32]
Infrared Imaging	Aflatoxin AFB_1_	Surface deposition from commercial samples	Stationary	Visible, near-infrared hyperspectral imaging	FDA, PCA	96%–100% validation accuracy for classification at 5 thresholds: 0, 10, 20, 100, 500 ppb.	[34]
Infrared Imaging/Fluorescence	Total Aflatoxin	Wound Inoculation	In motion	Infrared, Visible, and Ultraviolet Reflectance	RF	86% sensitivity and 97% specificity at a classification threshold of 20 ppb. Spectral data highly skewed.	[1]
Infrared Imaging	Total Aflatoxin	Artificial Inoculation	Stationary	Visible, near- infrared reflectance	PLS-DA	87% accuracy for classification of contaminated kernels at threshold of 20 ppb and 100 ppb.	[17]

* Acronyms definition: K nearest neighbors (KNN), Principal component analysis (PCA), Least square support vector machine (LS-SVM), Normalized Fluorescence index (NDFI), Partial Least-square principal component analysis (PLS-DA), Full with half maximum (FWHM), Partial least squares (PLS), Range factor (RF), Support vector machine (SVM), Factorial discriminant analysis (FDA).

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
