# Peer review of "A Review of the Methodology of Analyzing Aflatoxin and Fumonisin in Single Corn Kernels and the Potential Impacts of These Methods on Food Security"

_foods, 2020, doi:10.3390/foods9030297_

Round 1

Reviewer 1 Report

This manuscript is a review article covering methodology for the analysis for aflatoxin and fumonisin mycotoxins using a single kernel approach rather than bulk sampling. The argument for this is that it provides a less skewed picture of contamination since bulk samples, depending on the sampling procedure, do not represent the overall contamination in a batch of corn. Unlike other mycotoxins, the co-occurrence of these toxins, AFB1 and FB1, particularly in corn, is quite high and analysis at a single kernel level would offer improved mediation methods (such as sorting) that could impact food security. The topic of the review is somewhat limited compared to the methodology for the analysis of mycotoxins in general, but would be of use to researchers of fumonisin and aflatoxin, particularly in countries where corn is the staple cereal. The review is quite short, since literature on this topic is limited but has managed to cover 58 references on the topic under three general methodology protocols. There are no figures but Table 1 summarizes the analytical methods used, measurement type, corn source and accuracy from several of the references quoted. As a review, there is no experimental protocol per se to be considered nor the validity of data analysis.

It is generally well written and comprehensive, with a few typos as noted below.

Specific comments:

Page 2, line 61: Is the review paper referred to here Reference 1 or the manuscript itself, in which case it should read “this review paper”.

Page 2, lines 67-70: Either there are a couple of words missing from these two sentences or the grammar is not good. Maybe, “Secondly, this paper (we) will review……. Finally we will discuss….. “ or something similar.

Page 3, line 134: “mycotoxins” should be plural

Page 4, lines 171-172: There is something wrong with the grammar of this sentence

Page 7, lines 296-296: Is there a reference to this study?

Page 8, line 373: “nulk”? Do you mean “bulk”?

References: In several references, the authors are written “et al.”. I am not sure if this is acceptable to the journal or if all the authors need to be listed.

Reference 14: “Fusarium” is misspelled.

Reference 17: The final author appears to be missing.

Author Response

This manuscript is a review article covering methodology for the analysis for aflatoxin and fumonisin mycotoxins using a single kernel approach rather than bulk sampling. The argument for this is that it provides a less skewed picture of contamination since bulk samples, depending on the sampling procedure, do not represent the overall contamination in a batch of corn. Unlike other mycotoxins, the co-occurrence of these toxins, AFB1 and FB1, particularly in corn, is quite high and analysis at a single kernel level would offer improved mediation methods (such as sorting) that could impact food security. The topic of the review is somewhat limited compared to the methodology for the analysis of mycotoxins in general, but would be of use to researchers of fumonisin and aflatoxin, particularly in countries where corn is the staple cereal. The review is quite short, since literature on this topic is limited but has managed to cover 58 references on the topic under three general methodology protocols. There are no figures but Table 1 summarizes the analytical methods used, measurement type, corn source and accuracy from several of the references quoted. As a review, there is no experimental protocol per se to be considered nor the validity of data analysis.

It is generally well written and comprehensive, with a few typos as noted below.

Specific comments:

Page 2, line 61: Is the review paper referred to here Reference 1 or the manuscript itself, in which case it should read “this review paper”.

We have clarified the sentence by writing “this review paper,” which is currently at line 64, page 2.

Page 2, lines 67-70: Either there are a couple of words missing from these two sentences or the grammar is not good. Maybe, “Secondly, this paper (we) will review……. Finally we will discuss….. “ or something similar.

We have added the proper grammar wording to specify that we will be discussing about the review paper, which currently are at lines 69-71, page 2.

Page 3, line 134: “mycotoxins” should be plural

We made the word plural, currently at line 146, page 4.

Page 4, lines 171-172: There is something wrong with the grammar of this sentence

We corrected the grammar and wording of the sentences, currently at lines 184-185, page 5.

Page 7, lines 296-296: Is there a reference to this study?

We added the proper citation that shows the results, currently in lines 308-309, page 7.

Page 8, line 373: “nulk”? Do you mean “bulk”?

We corrected to “bulk”, line 384, page 9.

References: In several references, the authors are written “et al.”. I am not sure if this is acceptable to the journal or if all the authors need to be listed.

We changed bibliography format in Endnote to list all author names.

Reference 14: “Fusarium” is misspelled.

We corrected the misspelled word, currently reference 17.

Reference 17: The final author appears to be missing.

We changed bibliography format in Endnote to show all authors according to the format.

Reviewer 2 Report

The article entitled " A review of the methodology of analyzing aflatoxin and fumonisin in single corn kernels and the potential impacts of these methods on food security " pretend to describe the methods to analyze aflatoxin and fumonisin in single corn kernels, both historic methods and current techniques including liquid chromatography, fluorescence imaging, and infrared imaging.

The paper is well organized and presents essential and necessary data from a scientific point of view.

The main reason for the major revision are based in a review of aflatoxins and fumonisins determination methods should include the ELISA technique.

In addition, the methods for different aflatoxins and/or fumonisins can be detailed. I’ts not clear that all methodologies are for AB1 and for what fumonisin?

The second objective of the review, “then, review basic scientific findings, including classification and correlation of aflatoxin and fumonisin” are not clear.

 In detail:

1)     It is necessary to abbreviate the words aflatoxin (aflatoxins) and fumonisin (fumonisins) in the text.

2)     It should be detailed exactly what aflatoxins and/or fumonisis are determined.

3)     AFB1 appears but we don't know exactly what it is.

Author Response

The article entitled " A review of the methodology of analyzing aflatoxin and fumonisin in single corn kernels and the potential impacts of these methods on food security " pretend to describe the methods to analyze aflatoxin and fumonisin in single corn kernels, both historic methods and current techniques including liquid chromatography, fluorescence imaging, and infrared imaging.

The paper is well organized and presents essential and necessary data from a scientific point of view.

The main reason for the major revision are based in a review of aflatoxins and fumonisins determination methods should include the ELISA technique.

We added a new section, section 2.1, that explains the use of ELISA methods in mycotoxin analysis and its utilization in single kernel analysis of aflatoxin and fumonisin, currently located in lines 99-108, page 3.

In addition, the methods for different aflatoxins and/or fumonisins can be detailed. I’ts not clear that all methodologies are for AB1 and for what fumonisin?

We have address this problem, we specified the methods used in each experiment that was citated by showing the exact type of mycotoxin tested in sections 2.1, 2.2, 2.3, 2.4, 3.1, 3.2 and Table 1.

The second objective of the review, “then, review basic scientific findings, including classification and correlation of aflatoxin and fumonisin” are not clear.

We corrected the wording to be specific on the purpose of the review paper, currently in lines 69-71, page 2.

 In detail:

  • It is necessary to abbreviate the words aflatoxin (aflatoxins) and fumonisin (fumonisins) in the text.

We abbreviated each type of aflatoxin according to the method utilized in each citation. In addition, the words aflatoxin and fumonisin were abbreviated in each section of the review paper.

  • It should be detailed exactly what aflatoxins and/or fumonisis are determined.

We have specified in each citation the type of mycotoxin tested and clarified if they are testing total aflatoxin or fumonisin concentration or a specific type of aflatoxin or fumonisin.

  • AFB1 appears but we don't know exactly what it is.

We added citations to the introduction to explain the types of aflatoxin, currently lines 33-34, page 1, and types of fumonisin, currently lines 38-40, page 1, that are associated with mycotoxin contamination in corn.

Round 2

Reviewer 2 Report

The authors have followed the recommendations requested.
The article is suitable for publication